# Long Non-Coding RNAs as Determinants of Thyroid Cancer Phenotypes: Investigating Differential Gene Expression Patterns and Novel Biomarker Discovery

**DOI:** 10.3390/biology13050304

**Published:** 2024-04-27

**Authors:** Nicole R. DeSouza, Tara Jarboe, Michelle Carnazza, Danielle Quaranto, Humayun K. Islam, Raj K. Tiwari, Jan Geliebter

**Affiliations:** 1Department of Pathology, Microbiology and Immunology, New York Medical College, Valhalla, NY 10595, USA; ndesouza@student.nymc.edu (N.R.D.); tseymour@student.touro.edu (T.J.); humayun_islam@nymc.edu (H.K.I.); raj_tiwari@nymc.edu (R.K.T.); 2Department of Otolaryngology, New York Medical College, Valhalla, NY 10595, USA

**Keywords:** thyroid cancer, anaplastic thyroid cancer, papillary thyroid cancer, lncRNA, miRNA, cancer, transcriptomics

## Abstract

**Simple Summary:**

Thyroid cancer is the most diagnosed endocrine cancer. There are several types of thyroid cancer, which vary in aggressiveness and survival rate. Well-differentiated thyroid cancers, such as papillary and follicular thyroid cancer, are therapeutic-responsive, with high survival rates. Anaplastic thyroid cancer is undifferentiated, represents <2% of thyroid cancer cases, and is completely unresponsive to therapy, with a grim 5-year survival rate of <4%. Varying gene expression between non-cancerous and cancerous tissue may help define similarities and differences between these thyroid cancer types and, thus, help with the development of more effective therapeutic and diagnostic strategies. Long non-coding RNA molecules are identified as important cell regulators that can alter cell behavior at several different levels. The roles of several long-noncoding RNAs have been identified to characterize thyroid cancer. This review covers several long non-coding RNAs identified for their pathologic expression patterns and functions as cancer-promoting molecules.

**Abstract:**

Thyroid Cancer (TC) is the most common endocrine malignancy, with increasing incidence globally. Papillary thyroid cancer (PTC), a differentiated form of TC, accounts for approximately 90% of TC and occurs predominantly in women of childbearing age. Although responsive to current treatments, recurrence of PTC by middle age is common and is much more refractive to treatment. Undifferentiated TC, particularly anaplastic thyroid cancer (ATC), is the most aggressive TC subtype, characterized by it being resistant and unresponsive to all therapeutic and surgical interventions. Further, ATC is one of the most aggressive and lethal malignancies across all cancer types. Despite the differences in therapeutic needs in differentiated vs. undifferentiated TC subtypes, there is a critical unmet need for the identification of molecular biomarkers that can aid in early diagnosis, prognosis, and actionable therapeutic targets for intervention. Advances in the field of cancer genomics have enabled for the elucidation of differential gene expression patterns between tumors and healthy tissue. A novel category of molecules, known as non-coding RNAs, can themselves be differentially expressed, and extensively contribute to the up- and downregulation of protein coding genes, serving as master orchestrators of regulated and dysregulated gene expression patterns. These non-coding RNAs have been identified for their roles in driving carcinogenic patterns at various stages of tumor development and have become attractive targets for study. The identification of specific genes that are differentially expressed can give insight into mechanisms that drive carcinogenic patterns, filling the gaps of deciphering molecular and cellular processes that modulate TC subtypes, outside of well-known driver mutations.

## 1. Introduction

Thyroid cancer (TC) is the most common endocrine malignancy, with increasing incidence worldwide [1]. Increasing incidence is attributed, in part, to successes in early tumor detection (i.e., increased tumor surveillance and diagnostic efficiency), and environmental exposure to iodine levels, as well as physiological factors that impact endocrine function (i.e., diet, obesity, genetics) [2]. Benign thyroid nodules are common clinical observations but are subject to close monitoring to ensure there is no malignancy developed. TC represents all malignant nodules, which are diagnosed typically using thyroid fine-needle aspiration (FNA) [3]. Further molecular testing in addition to FNA has aided in differentiating TC subtypes. TC is divided into several subcategories based on degree of differentiation—placing these subtypes into vastly different treatment and survival brackets [4]. Well-differentiated TC includes papillary thyroid cancer (PTC) and follicular thyroid cancer (FTC), both of which are relatively responsive to traditional therapeutic intervention and have comparably higher survival rates. Undifferentiated TC includes anaplastic thyroid cancer (ATC), which is one of the most aggressive malignancies, characterized by its fast-growing pace and refractoriness to therapeutic modalities. Amongst the several TC subtypes, PTC and ATC represent the broadest range of thyroid follicular cell differentiation [5]. Molecular descriptions of these carcinogenic mechanisms are indebted to the field of cancer genomics; genomics has served as a critical foundation of comprehensive genomic evaluation. Cancer is driven by both common and unique orchestrations of genomic events; the selective malignant transformations of normal cells is a direct result of favorable genomic alterations. These alterations impact the cell even further on the transcriptomic, proteomic and epigenomic level. Differential gene expression patterns that propagate and drive carcinogenic behavior has enabled extensive investigation of these cellular established molecular programs [6]. Gene expression characterization and the cellular role of these TC subtypes is a critical need for establishing a better understanding of the molecular mechanisms that shape these subtypes, as well as the mechanisms that make each subtype unique.

There are several well-classified mutations that are involved in the initiation and progression of TC. Most TC cases present, genetically, with driver mutations in the MAPK (mitogen activated protein kinase) pathway. Mutations in this pathway include primarily mutated BRAF and RAS, significantly contributing to the constitutive activation of this highly proliferative signal transduction cascade [7]. Other common mutations exist in the p53 tumor suppressor, bypassing and removing regulatory cell-cycle control “brakes”. Other genomic abnormalities seen in TC include the RET/PTC (RET/PTC1, RET/PTC3) rearrangement, which involves the fusion of the RET tyrosine kinase domain with genes expressed in thyroid follicular cells, supplying a constitutively active promoter for RET tyrosine kinase activity [8]. Additionally, PAX8/PPARγ rearrangements have been associated with a more aggressive PTC phenotype [9]. Despite the identification of these driver mutations, they are deemed inefficient as prognostic factors for TC; thus, mutational study alone is insufficient for prognostic, diagnostic, or therapeutic development, thus highlighting the need for additional molecular evaluation [7].

Long non-coding RNA (lncRNA) molecules are non-protein coding transcripts that exceed 200 nucleotides (nt) in length and possess a wide variety of functions in both physiological and pathological cell and tissue conditions [10]. Functionally, lncRNAs can regulate gene expression at the genomic, transcriptomic, and proteomic levels, either through the molecular “sponging” effect or through direct interaction via extensive nucleic acid and protein-binding domains. Small ncRNAs known as microRNAs (miRNAs) are 19–25 nt long, and function as negative regulators of messenger RNAs (mRNAs) via regions of sequence complementarity and direct binding. lncRNAs also possess these RNA-binding domains and have the ability to bind and “sponge” miRNAs, thus acting as direct negative regulators of miRNAs as well as indirect positive regulators of mRNAs. This role is identified as a competing endogenous (ce) network of transcriptomic interactions [11]. The expression patterns of lncRNA molecules extensively directs the fate of these epigenetic interactions, and the role of these RNA molecules will determine the functional consequence of cellular activity [12]. lncRNA molecules have been marked as central molecular regulators in a multitude of cancer types. Gene expression analysis of tumors have given way to the identification of both novel lncRNAs and previously unknown functions of known lncRNAs. The dysregulated expression patterns of lncRNAs contribute immensely to shaping carcinogenic programs and the development of treatment resistance—driving cancer cell survival [13].

Molecular biomarker identification is a critical avenue of both diagnostic and therapeutic intervention. The concept of cancer heterogeneity lies in the premise that all cancers vary, and the same cancer can vary both case-by-case and vary in primary and secondary tumor sites, as well as within tumor sites. Thus, the heterogeneity of tumors is a significant representation of the diagnostic and therapeutic hurdles faced [13]. Identification of molecular signatures that contribute uniformly among cancers is a significant gateway into rapid diagnosis and a more accurate prognosis. The dynamic nature of cancer onset and progression is orchestrated by a series of molecular changes that drive its unique behavior; identification and evaluation of biomarkers contributes significantly to understanding the cellular behavior of cancer cells in isolation and in concert [14,15]. These biomarkers could fall under several categories—they can be genomic, transcriptomic, or epigenomic components that function to regulate or direct carcinogenic transformation [16]. lncRNA molecules function at the interface of intracellular networks; their ability to modulate genomic readouts at the DNA, RNA, and protein levels makes them indispensable candidates for evaluation. The literature has reported a plethora of lncRNA molecules that have become candidates of functional studies—their dysregulated expression patterns have brought these molecules a significant amount of research attention, which in turn, has confirmed their significant contribution to carcinogenic establishment [17].

## 2. Anaplastic Thyroid Cancer

ATC is the most aggressive and lethal endocrine malignancy. ATC’s 5-year survival rate is less than 4% when metastatic, with an overall predicted survival of 4–12 months upon diagnosis [18]. Reflecting this poor survival, ATC has a unique staging criterion, as it is always diagnosed as stage IV metastatic disease (i.e., stage IVA, IVB, IVC). Stage IVA represents 10% of ATC cases that contain cancer localized to the thyroid. Stage IVB represents 40% of ATC cases that contain cancer that has spread from the thyroid to regions of the neck. Most ATC cases (>50%) are represented by a stage IVC diagnosis, where the cancer has metastasized to distant organs, primarily targeting the brain and the bones, eliminating efficacy of most treatment options [18]. Rapid identification of ATC is a critical hurdle to overcome—due to its highly dedifferentiated and fast-growing nature, it is extremely difficult to detect and harness at earlier stages [5]. Less than 20% of patients with ATC are alive one year after diagnosis [19,20], highlighting the need for molecular exploration and evaluation.

## 3. Differentiated Thyroid Cancer

Well-differentiated thyroid cancers (WDTC) arise from the follicular cells of the thyroid and consist of two subtypes—PTC and FTC. Disease-related mortality 10 years following WDTC diagnosis is lower than 5% [21]. Collectively, WDTC is slow-growing, presents with feasible early detection, and high therapeutic responses. However, aggressive cases of WDTC do arise and distant metastases significantly increases mortality rate [22]. This paradox of disease-related mortality interestingly emphasizes the value of biomarker identification in cases with high survival, as well as those with aggressive disease [23].

### 3.1. Papillary Thyroid Cancer

PTC accounts for approximately 80% of all TC cases and has been increasing in incidence over the last few decades [10]. PTC disproportionately affects women (2.6 times higher incidence in women [22]), being the seventh most common cancer in women in the United States and one of the most common cancers in women under 25 years old [23]. Even though PTC is more common in women, it is a more aggressive disease in men [24]. Factors that may contribute to this sex-related discrepancy may be a result of more frequent screening processes in women, or the fact that PTC is more commonly diagnosed in women when they are younger in age compared to male PTC diagnoses, therefore causing more favorable age-related outcomes [25]. PTC is often detected early and is curable, contributing to its high five-year survival rate [26]. Since PTC predominantly affects a younger population, a high 20-year survival rate does not imply that the patient will reach an average life expectancy without recurrent disease. Lymph node (LN) metastases are frequently present at diagnosis (50–60%) and due to treatment, do not worsen the prognosis [27]. However, as with all cancers, a major concern in PTC is metastatic spread and invasion into the surrounding tissue [28]. PTC diagnosis and staging are critical to determine prognosis, therapeutic strategies, and intensity of follow-up needs. ATC can exist alongside well-differentiated PTC; however, most cases occur spontaneously, without previous or simultaneous PTC pathologies [4]. Analysis of molecular profiles can hence serve as promising biomarkers or actionable therapeutic targets.

### 3.2. Follicular Thyroid Cancer

FTC is the second most common form of TC [29] and is most frequently diagnosed in patients 45–54 years of age [30]. Despite its shared follicular cell differentiation status, FTC varies from PTC and is significantly more difficult to diagnose, leading to many instances of overtreatment. Difficulty in diagnosis lies in the inability to effectively distinguish amongst the varied follicular patterned lesions on the thyroid from a bona fide FTC tumor. Efficient diagnostic strategies for PTC, such as FNA, are not well-suited for FTC, placing a significant need for additional diagnostic strategies [31]. The presence of vascular invasion at the time of diagnosis, accompanied by the absence of typical PTC nuclear features, distinguishes FTC phenotypically from PTC. Thus, FTC is further subdivided into minimally invasive FTC (MI-FTC) and widely invasive FTC (WI-FTC) [32]. Additionally, compared to PTC, FTC has a higher metastatic propensity; spread to distant sites (i.e., lungs and bone) at diagnosis is more frequently observed (7–23% of cases) and is a significant prognostic predictor of survival [33]. In the absence of metastatic spread, the overall 5-year survival rate of FTC is almost 100% when treated (roughly 98%); however, poorer prognostic outcomes increase with age. With more aggressive FTC, the 10-year survival rate is approximately 77%, and 20-year survival is 33.7% [31]. This age-related risk is a direct result of increased probability of osteolytic bone metastasis and lung disease. Standard treatment for FTC includes total thyroidectomy, and if minimal spread is present, local neck dissection. In instances where surgical resection is not plausible, treatment with radiation therapy is standard for locally advanced disease for containment in adjacent structures [30]. As stated, FTC is difficult to detect via differential diagnoses, and despite its differentiated classification, it is typically accompanied with metastatic spread and a lack of early detection capabilities. Thus, the identification of molecular biomarkers that can aid in the differential diagnosis of FTC or aid in the prognosis of FTC patients are critical unmet needs.

## 4. Anaplastic Thyroid Cancer vs. Well-Differentiated Thyroid Cancers (Papillary Thyroid Cancer and Follicular Thyroid Cancer)

Although ATC, PTC, and FTC are types of thyroid carcinoma, expression patterns and genomic drivers are often distinct and may play extensively differing roles in these TC subtypes. For example, gene expression analysis of ATC vs. PTC patient tissues using Gene Expression Omnibus datasets (i.e., GSE33630, GSE3678) analyzed with GEO2R software confirms there are genes that are statistically differentially expressed in ATC compared to PTC [34]. Thus, each thyroid carcinoma must be distinctly classified, but may subsequently give way to understanding why certain cases are driven in either more or less aggressive directions. Further, some evidence supports the transition from a differentiated form of TC into aggressive, undifferentiated ATC [35].

With the slow-growing nature of PTC, and fast-growing nature of ATC, it is extremely valuable to understand the molecular similarities and differences that exist between subtypes, which could lead to beneficial insight and discovery for thyroid carcinoma subtypes [36].

## 5. Current Thyroid Cancer Treatment Modalities

As indicated above, treatment plans for ATC and PTC vary significantly due to the vast differences in growth rates and staging criteria. ATC stage IV diagnostic criteria centers on the premise that all cases are diagnosed as stage IV metastatic disease. Treatment of ATC highly depends on stage IVA, IVB, IVC status, with decreased options and efficacy, successively [37]. Thus, most ATC patients are prescribed elements of palliative care [38]. Selection of the small molecule inhibitor to use as combination therapy is tailored specifically to the genetic profile of the patient’s tumor. This profiling, generally, focuses on the BRAF status of the patient [39], highlighting the need for the identification of other markers of treatment evaluation and outcome. PTC treatment typically begins with surgical resection at time of diagnosis. Either partial or complete removal of the thyroid gland (lobectomy or thyroidectomy, respectively) is warranted, depending on the size and stage of the tumor or tumor development. The slow-growing nature of PTC makes surgical resection a plausible option, unlike ATC, where its rapid growth makes surgical intervention almost impossible. More severe cases of PTC do arise; thus, total thyroidectomy followed by radioactive iodine therapy aids in the prevention of a more severe disease recurrence. Total thyroidectomy, however, varies from removal of the thyroid completely, or removal of the thyroid as well as the extracapsular (extrathyroidal) region. Extrathyroidal extension (ETE) is associated with increased risk of recurrence, thus extensively contributing to disease prognosis [40]. Despite the high percentage of survival outcomes in PTC, extensive molecular analysis of the varied aggressiveness and instances of recurrence is warranted. In both instances of ATC and PTC, functional annotation of transcriptomic profiles can aid in disease evaluation and curative efforts.

## 6. Studying lncRNAs

Progress has been made in recent years in methods and technology developed to study transcriptomic elements. Development of high-throughput databases has contributed immensely to the study and identification of noncoding RNAs across cancer types. Gene Expression Omnibus (GEO), a publicly available database provided by National Center for Biotechnology Information (NCBI), enables evaluation of patient genomic profiles through a fold-change analysis using GEO2R software (accessed on 15 April 2024, https://www.ncbi.nlm.nih.gov/geo/). The Cancer Genome Atlas (TCGA) serves as an extensive source of genomic data obtained from greater than 20,000 primary cancer and matched normal patient samples (accessed on 11 April 2024, https://www.cancer.gov/tcga). The Atlas of non-coding RNA in cancer (TANRIC) provides expression profiles of patient data, providing a source for the identification and evaluation of lncRNAs across 20 cancer types [41]. RNA–RNA interactions are largely regulated by hybridization energies and sequence complementarity—making algorithmic prediction software packages high yield for successful functional analyses [42,43,44,45,46]. Some software examples include lncRNA2Target [47,48], DIANA-lncBase v3 [49], and IntaRNA [50,51,52,53]. For in vitro assessment of select ncRNAs, qRT-PCR and RNA sequencing are widely used for expression identification and comparative assessment [54]. Transcriptomic silencing methods such as RNA-interference, as well as gene editing technologies such as CRISPR, have enabled functional analyses that can be translated beyond in vitro evaluation [55,56].

## 7. lncRNAs in Anaplastic Thyroid Cancer

The molecular characterization of ATC has made tremendous progress in the last few years; however, prognosis, evaluation, and treatment response still remain prominent issues [57]. As such, despite its rapid growth and highly metastatic nature, mutational burden is not a significant contributor to genomic instability [58]. Further, common driver mutations of ATC are shared with well-differentiated PTC, suggesting there are likely other biological factors that orchestrate the varied pathologies of these two TC subtypes. This premise highlights the role of epigenetics; identification of factors that fine-tune genomic expression may give insight into the molecular mechanisms that cause aggressive ATC cell phenotypes [59].

The need for biomarker identification in ATC lies at various levels of intervention; molecules that describe its unique, fast-growing nature, its resistance to conventional and innovative therapeutics, and its grim patient survival outcomes will enable a greater understanding of why ATC is so difficult to characterize. Due to the therapeutic refractive nature of ATC, identification of molecules that can serve as prognostic or diagnostic biomarkers is a critical need aimed at improving patient outcomes [60]. Analysis of differentially expressed lncRNAs in ATC tissue have contributed to the annotation of both the genomic integrity and cellular functions of this aggressive cancer type. The following lncRNAs discussed in this review have been functionally annotated in ATC and have contributed to the understanding, in part, of how its carcinogenic programs are established. Both the in vitro and in vivo experimental designs of lncRNA assessment in the following studies were evaluated according to the genomic profiling of patient tissues. Assessment of functional consequences of the over- or under-expression of candidate lncRNA molecules in vitro and in vivo was accomplished using gene-editing technologies (Table 1).

H19

Raveh et al. previously reviewed the several roles of lncRNA H19 in cancer initiation, progression, and metastasis. Physiologically, H19 is expressed initially in the cells of a developing embryo, and its expression is subsequently downregulated at birth. After birth, H19 is exclusively expressed in tumors. H19 specifically has been denoted for its role in promoting cancer cell survival in the most adverse conditions; tumors with high H19 profiles substantially withstand hypoxic conditions and evade cell death pathways. These mechanisms of action gave H19 a role in the resistance of several drug treatments [61]. Previous works have shown that H19 expression can be induced by EMT-agonists. In hepatocellular carcinoma, H19 expression was induced with TFG-β as a readout of the PI3K/Akt signaling pathway [62]. As stated, ATC cell proliferation is driven, in part, by mutations in the MAPK pathway, resulting in constitutively active signaling. Ras, a downstream regulator of the MAPK pathway, can also bind to PI3K, leading to subsequent activation of this pathway [63].

A study conducted by Zhang et al. reported a gene expression analysis of 19 ATC tissue samples compared to 19 benign thyroid nodule (BTN) samples, evaluating differentially expressed transcripts between these groups. H19 was identified as a differentially expressed transcript, exhibiting a significantly increased expression in ATC tissue compared to the BTN tissue [64]. Upon in vitro assessment of H19 function, it was found that this molecule had a significant impact on proliferation, colony forming capacity, invasive potential, migratory propensity, and apoptotic induction in ATC cell lines. Further in vivo assessment confirmed a role of H19 in ATC tumorigenesis, which was evaluated through a reduction in tumor volume and lung metastasis when H19 expression was reduced [65]. As stated, the physiological expression of H19 is low in bodily tissues following embryonic development, so much so, its expression is often never detected in non-cancerous tissue [66]. It is suggested that this phenomenon may be a contributing factor as to why H19 has contradicting roles in ATC vs. PTC, a concept that will be further discussed in the context of PTC, below.

MALAT1

LncRNA MALAT1 (Metastasis Associated Lung Adenocarcinoma Transcript 1) has been reported previously for its function in both cancer development and progression, playing critical roles in central carcinogenic mechanisms (i.e., proliferation, apoptosis, tumorigenicity, etc.). Li et al. reported that MALAT1 has significant impact on tumor size, location of the tumor, and the cancer stage, highlighting a potential biomarker role [67]. MALAT1 is generally proposed to function through the activation of the MAPK and the PI3K/Akt pathway [68]. MALAT1 exerts its oncogenic function through the inhibition of miR-124, resulting in the subsequent upregulation of Slug in lung adenocarcinoma [69]. Further, Qin et al. reported a RP11-395G23.3/miR-124-3p/ROR1 ceRNA axis in ATC. In this case, miR-124-3p functions as a negative regulator of ROR1 in ATC. lncRNA RP11-395 is upregulated in ATC patient tissues in the dataset provided by GEO (GSE33630) [70]. This premise exemplifies the concept of several different lncRNAs affecting the outcome of a single miRNA molecule’s regulatory role. Both instances led to aberrant proliferative signaling.

Gou et al. also reported MALAT1 as a differentially expressed transcript in ATC patient tissue. Thirty ATC tissues were compared to normal adjacent thyroid tissue samples, and MALAT1 was identified as a significantly upregulated transcript in ATC. The in vitro assessment of MALAT1 confirmed a role for this molecule in ATC cell proliferation, migration, invasion, autophagy, and apoptotic induction. This study thus further evaluated the molecular innervations of MALAT1 that give this molecule a significant role in the crosstalk of these central carcinogenic mechanisms. Gou et al. identified MALAT1 as a ceRNA, through sequence complementarity and thus direct interaction with miR-200a-3p. Further evaluation confirmed a role for miR-200a-3p as a tumor suppressor and is functionally associated with increased apoptosis and decreased migration and invasion of ATC cells. The aforementioned ceRNA network suggests a need for further evaluation of the mRNA targets of miR-200a-3p. Further in vitro evaluation of predicted interactions between miR-200a-3p and FOXA1 (Forkhead Box A1) confirmed that the tumor-suppressive effects of miR-200p-3a are mediated through negative regulation of FOXA1. FOXA1 contributes to a multitude of pathological transformations, promoting proliferation and cellular motility in other carcinomas. Taken together, the aberrant expression of MALAT1 in ATC leads to a decrease in miR-200p-3a expression, and thus, a dysregulated increase in FOXA1 translation, which contributes, in part, to the progression of acquired carcinogenic transformation [71].

HOTAIRM1

An additional study of ceRNA networks in ATC was conducted by Zhang et al., identifying and evaluating the interaction of HOTAIRM1 (HOXA Transcript Antisense RNA) and miR-144. HOTAIRM1 was selected as a candidate for study due to its high expression correlated with decreased ATC patient survival. This ceRNA network is unique; HOTAIRM1 was identified to bind primary miR-144 (pri-miR-144), preventing its well-described functional miRNA processing interaction with DROSHA—blocking successive maturation and production of miR-144. The functional role of mature miR-144 blocks epithelial-to-mesenchymal (EMT) transition of ATC cells, thus, its decreased expression as a result of overexpressed HOTAIRM1 significantly drives this cellular phenotypic transformation [72].

UCA1

The functional role of cellular Myc (c-Myc) has been well classified in many cancer types as an overexpressed oncogene that drives cancer cell progression and survival. c-Myc significantly promotes cell growth and proliferation through the activation of genes involved in biosynthetic pathways of macromolecules and cell cycle genes. c-Myc is relatively well studied in many carcinomas since the time of its initial identification for having extensive pro-tumorigenic functions. Additionally, c-Myc plays transcriptomic regulatory roles through the inhibition of miRNA expression in a highly selective fashion—i.e., blocking miRNAs that function as tumor suppressors, thereby also supporting its role as an oncogene [73]. A study had identified lncRNA UCA1 (urothelial carcinoma-associated 1) as a potential regulator of c-myc. Initial identification of UCA1 was a result of its significant upregulation in ATC patient tissue. Further evaluation of UCA1 in ATC cell lines enabled identification of miR-135a, a miRNA molecule that UCA1 competes with c-myc for binding. Thus, dysregulated overexpression of UCA1 contributes to the aberrant expression of c-myc, driving its oncogenic behavior. Wang et al. further demonstrated in vitro that miR-135a inhibition upregulates c-myc expression, driving ATC cell proliferation [74].

An additional study of UCA1 in ATC described its role in the reduction in CD8+ T lymphocyte cell cytotoxic activity. The effect of lncRNA molecule expression on the activity and action of T lymphocytes and other immune cells are documented in several cancer types. ATC, specifically, has a unique immune cell population that drastically shapes its tumor microenvironment (TME). Wang et al. reported a critical interactive network that exists between UCA1 and miR-148a, by which miR-148a, in turn, negatively regulates the expression of programmed death ligand 1 (PD-L1). PD-L1 is a crucial, cell surface-expressed immune regulatory molecule that contributes to the shutdown of immune cell activation when bound to its cognate receptor, programmed death-1 (PD-1). Therefore, high UCA1 expression leads to high PD-L1 expression through the “sponging” of miR-148a, resulting in the increased shut down of cytotoxic T lymphocytes (CTLs), attenuating their anti-cancer targeting efficiency [75]. A retrospective study of ATC patients reported that high expression of PD-1/PD-L1 proteins in ATC tumor samples was directly correlated to, and served as a predictive marker of progression-free survival and overall survival (OS) in ATC patients that received multimodal therapy [76]. Thus, results reported by Wang et al. confirms a potential lncRNA-related biological mechanism that affects PD-1 and PD-L1 expression in ATC.

MANCR

Huang et al. evaluated differentially expressed transcriptomic molecules in ATC vs. normal thyroid samples using whole transcriptome sequencing. Upon analysis, MANCR (mitotically associated long non-coding RNA) was identified as an overexpressed transcript in ATC when compared to both normal thyroid tissue and PTC samples. To evaluate the potential function of MANCR in these patient data, in vitro experimental design in ATC cell lines confirmed a role for this molecule in apoptotic inhibition and both proliferative and invasive induction. Thus, Huang et al. confirmed MANCR as a potential ATC tumor promoter [77].

As mentioned, a significant issue when combating ATC is almost complete treatment resistance. Lack of chemotherapeutic response is characteristic of ATC; however, the exact molecular mechanisms that contribute to this resistance have not been fully elucidated. The identification of molecules that may be causing intracellular conditions that lead to ATC tissue treatment refraction is an invaluable feat. Studies are currently underway evaluating the use of RNA-interference (RNAi) technology to generate short-interfering RNAs (siRNA) against miRNA molecules that have been identified to confer cellular resistance to chemotherapeutic agents [78]. The success of these studies gives great insight into potential ways ATC tumorigenicity can become more amenable to common treatment modalities.

NEAT1

A study conducted by Yan et al. evaluated the role that lncRNA NEAT1 (nuclear paraspeckle assembly transcript 1) has on ATC chemoresistance. NEAT1 expression was found to be significantly increased in ATC patient tissue (26 tumor tissues vs. adjacent non-tumor tissue) and its expression was associated with advanced TNM stage and LN metastasis. In support of this finding, NEAT1 transcripts were also elevated in ATC cell lines, and its function was evaluated in vitro. NEAT1 was found to contribute to central ATC carcinogenic mechanisms, exerted by a molecular interaction with miR-9-5p. miR-9-5p has been evaluated in other cancer types, having a role as a tumor suppressor, as well as a significant contributor to decreased resistance to cis-diamminedichloroplatinum(II) (DDP) treatment in ovarian cancer. This study presented the findings that miR-9-5p also made ATC more amenable to a chemotherapeutic response to DDP, thus placing the inhibitory role of NEAT1 on miR-9-5p as a significant contributor to DDP resistance. The ceRNA axis also places value on the mRNA targets of miR-9-5p. SPAG9 (sperm-associated antigen 9) is an identified target of miR-9-5p, and has been shown to enhance Taxol resistance in breast cancer. To this end, SPAG9 transcripts were also elevated in the same ATC patient dataset used by Yan et al., and SPAG9 expression was positively correlated with NEAT1, and inversely correlated with miR-9-5p, supporting this proposed regulatory network. Yan et al. proposed that the silencing of NEAT1 expression in ATC may decrease the resistance to DDP through this regulatory network, marking a tremendous contribution to understanding how ATC cells may be conferring this robust treatment resistance [79].

Conditions of hypoxia are prominent issues in rapid growing solid tumors (such as ATC) and it is deemed a significant contributing factor to chemotherapeutic and treatment resistance. Powell et al. evaluated the effect of common hypoxia-responsive miRNAs, such as miR-210-3p in ATC, and found that this miRNA has the potential to serve as a hypoxic marker [80]. Further, an additional study evaluating NEAT1 described its overexpression as a key driver of hypoxia-induced carcinogenic behavior in ATC. In support of Yan et al., this study also observed an overexpression of NEAT1 in ATC tissues and cells and evaluated its role in vitro and in vivo. It was concluded that NEAT1 negatively regulates the expression of miR-206 and miR-599 through the “sponging” effect—which in turn, leads to an increase in migration, invasion, and glycolytic pathways in hypoxic conditions, supporting the growth and spread of ATC in the most adverse of conditions [81].

PAR5

Despite the commonality of studying overexpressed lncRNAs, underexpressed transcripts also contribute significantly to both understanding molecular mechanisms and establishing pathologies. The expression of the lncRNA, PAR5 (Prader Willi/Angelman region RNA5), is significantly decreased in ATC tissue samples compared to normal thyroid tissue. Interestingly, PAR5 is not amongst the differentially expressed genes in PTC. Pellecchia et al. reported that PAR5 directly interacts with EZH2, which prevents the transcription of E-cadherin [82]. E-cadherin and N-cadherin are cell adhesion molecules that also serve as markers of EMT. During EMT, a transforming cell will lose its epithelial-like phenotype and downregulate E-cadherin and gain a mesenchymal-like phenotype and upregulate N-cadherin [83]. Thus, a loss in E-cadherin expression can drive EMT processes and significantly contribute to metastatic spread [81]. Therefore, in ATC, low expressed PAR5 is unable to inhibit EZH2, which is well-known for its oncogenic activity in states of overexpression/ hyperactivity [82].

## 8. lncRNAs in Well-Differentiated Thyroid Cancer

### 8.1. Papillary Thyroid Cancer

The molecular characterization of PTC is a critical component of deciphering mechanisms of action and understanding how/why disease recurrence remains a prominent issue. Even though PTC has a high survival rate and is responsive to treatment and/or surgical resection, aggressiveness and metastasis are still plausible outcomes [24]. Additionally, PTC is the most common cancer diagnosed in women < 25 years of age, raising tremendous questions of how this specific subtype can arise early in life, when compared to the more aggressive ATC subtype, which is more commonly diagnosed at >60 years of age [20]. Both endocrine and non-endocrine related lncRNA molecule discovery and functional annotation are indispensable pieces of information in PTC study. The expression of lncRNAs SOX2OT (SOX2 overlapping transcript), DANCR (differentiation antagonizing non-coding RNA), and TINCR (tissue differentiation-induced non-coding RNA) were reported to serve as biomarkers in PTC, due to their association with various clinicopathological features. Upregulation of DANCR and SOX2OT in PTC was observed to correlate with cancer onset, and cancer onset and progression, respectively [83]. Further, the role of DANCR in PTC is poorly studied [84]. Of note, authors reported no significant change in expression of TINCR in PTC vs. normal tissue, contradicting previous reporting of its upregulation in the same disease [85]. This displays the complexity of lncRNA expression, as well as the varying consequences of their dysregulated expression patterns [86].

The following lncRNAs discussed in this review have been functionally annotated in PTC and have contributed to the understanding, in part, of how its carcinogenic programs are established and maintained. Both the in vitro and in vivo experimental designs of lncRNA assessment in the following studies were evaluated according to the genomic profiling of patient tissues. Evaluation of the functional consequences of over- or underexpression of candidate lncRNA molecules in vitro and in vivo were accomplished using gene-editing technologies (Table 1).

Lnc-OMD-1

As mentioned, instances of ETE in PTC are clinically associated with a more aggressive phenotype. The molecular characterization of events that promote ETE, or serve as a marker of ETE, is a critical avenue of study. Chen et al. evaluated the transcriptional regulatory network that may promote ETE in PTC cases using extensive bioinformatic analyses. Within this study, lncRNA lnc-OMD-1 expression was identified via in silico analysis of patient datasets, as being significantly correlated with ETE. The function and correlation of lnc-OMD-1 in/to PTC (or other cancers) has not yet been evaluated experimentally, but would be a beneficial feat [87].

LUCAT1

Luzon-Toro et al. proposed the role of lncRNA LUCAT1 (lung cancer associated transcript 1) as a novel prognostic factor in PTC patients. Evaluation of gene expression data from 61 PTC tissues vs. adjacent non-tumor tissue revealed LUCAT1 as the most significantly upregulated transcript, highly correlated with advanced TNM staging. It was found that LUCAT1 is expressed within the nucleus of PTC cells and is highly involved with cell cycle control (G1 phase). Further in vitro analysis of LUCAT1 places this lncRNA as a significant contributor to PTC cell proliferation, invasion, and apoptotic prevention. LUCAT1 involvement in these mechanisms were proposed to occur through p53 and p21, two proteins that are responsible for cell cycle arrest. The expression levels of p53 and p21 were inversely correlated with the expression of LUCAT1 in both the wild-type and knockdown model, suggesting a connection between these molecules and the ability for LUCAT1 to promote cell cycle arrest [88].

HOTTIP

The lncRNA HOTTIP (HOXA transcript at the distal tip) has been identified for its close association with increased pathologies in a multitude of cancer types. HOTTIP is located at the 5′-end of the Homeobox A genomic region, and has been found to regulate various cellular processes, functioning as a promoter for robust cellular proliferation and metastasis. Evaluation of clinical tissue samples aided in the identification of the HOTTIP-miR-637-Akt1 regulatory axis. HOTTIP was robustly expressed in PTC tissues and in vitro PTC cell lines, and directed carcinogenic behavior in culture through the negative regulation of miR-637 expression. miR-637 negatively regulates the expression of Akt1; thus, high HOTTIP levels enable aberrant Akt1 expression—driving proliferative and metastatic behavior. Additional in vivo assessment of this interactive axis identified a role for HOTTIP in PTC tumorigenesis—supported by a significant decrease in tumor volume when HOTTIP expression was transcriptionally repressed. Further, tumors with low HOTTIP expression had higher miR-637 and lower Akt1 levels, supporting the claim that carcinogenic and tumorigenic promotion was fine-tuned by this interconnected molecular interaction [89].

LINC00313

Further evidence of the impact that the ceRNA network has on promoting cancer cell progression, lncRNA LINC00313 (long intergenic noncoding RNA) was identified by Wu et al. as a significantly upregulated transcript in PTC tissue vs. normal tissue, and that this increase in expression was correlated with a worse prognosis. Upon in vitro assessment of LINC00313 function, this molecule was deemed a significant promoter of PTC cell migration and clonogenicity, whilst also having the capability to arrest apoptotic induction. LINC00313 was found to accomplish its oncogenic role through the downregulation of miR-4429. Rescue of miR-4429 expression was found to revert the oncogenic functions of LINC00313, functioning, in part, as a tumor suppressor [90].

H19

As mentioned, lncRNA H19 is an overexpressed transcript that contributes significantly to ATC tumorigenesis. However, in PTC, H19 is identified as an under expressed transcript when compared to normal thyroid tissue, and its decreased expression is correlated with promoting carcinogenic behavior [91]. Some studies [92] have identified the contrary, suggesting that H19 is overexpressed in PTC, and functions as an oncogene, similarly to its functional annotation in ATC. The dynamic and versatile functions of H19 within the same cancer type further drives the message of lncRNA complexity, and genomic heterogeneity from patient-to-patient and cohort-to-cohort. The expression levels of lncRNAs depend on an abundance of other upstream factors that will lead to their differential expression—making the understanding and evaluation of genomic readouts highly convoluted. In support of this variation, other factors, such as immune cell infiltrates, contributed extensively to the outcome of the tumor by shaping the TME. A study conducted proposed that the TME of PTC is modulated, in part, by low H19 levels. Co-expression evaluation of H19 and immune cell population proposed that H19 expression is positively correlated with B cells, T cells (CD4+ and CD8+), as well as dendritic cells in PTC. Therefore, H19 up- and downregulation may contribute to immune cell infiltration in the TME of PTC in a varied, and tumor-specific manner. The dysregulated expression of H19 in PTC, regardless of if it is higher or lower compared to normal tissue, has a negative impact on patient survival and is correlated with more severe disease in any differentially expressed event [93].

BANCR

A common driver mutation in TC subtypes is a mutant BRAF molecule, which leads to immortalized cell proliferation. BANCR (BRAF-activated non-coding RNA) has been shown to function as both a tumor suppressor as well as an oncogene. Liao et al. evaluated the expression levels of BANCR in PTC patient tissue vs. normal thyroid tissue and found that its expression was significantly downregulated in PTC. In addition, low BANCR levels were correlated with advanced TNM staging. Differentiation status of thyroid carcinoma is a central characterizing factor between subtypes. Well differentiated PTC was found to have higher BANCR levels (i.e., thus less severe case), and lower BANCR levels were correlated with a more poorly differentiated PTC in tissue samples. Evaluation in vitro enabled the conclusion that when BANCR is experimentally overexpressed, it inhibits proliferation and induces apoptosis in PTC cell lines and in vivo. The overexpression of BANCR in PTC cell lines that harbor the BRAFV6000E mutation were found to have inactivated ERK1/2 and p38, which are downstream readouts of RAF signaling. Use of a MAPK inhibitor (U0126) successfully blocked the inactivation of ERK1/2 and p38 in these cell lines. To support this, following BANCR overexpression in a PTC cell line lacking the BRAFV600E mutation (contains RET/PTC translocation), the aforementioned effect on ERK1/2 and p38 was not seen. This finding presents a potential therapeutic strategy for patients with PTC derived from BRAFV600E driver mutation and lower BANCR expression levels [94].

COMET

Esposito et al. identified lncRNA COMET (Correlated-to-MET) for having an oncogenic role in the development and progression of BRAF and RET/PTC genetic driven PTC. COMET is a natural antisense transcript (i.e., transcribed from the antisense strand of MET) that is highly expressed in PTC patients that harbor both genomic drivers. Further structural analysis of COMET revealed a unique transcriptional position that includes a transcription start site that is located within the first intron of MET. Functionally, COMET was found to activate components of the MAPK pathway. MET, a downstream regulator of the MAPK pathway, functions as an oncogene in a multitude of cancer types (TC, included). COMET expression was found to be statistically correlated with MET expression, and depletion of COMET resulted in a significant reduction in MET expression as well as other MAPK oncogenes. Treatment of immortalized normal thyroid cell line Nthy-ori-3-1 with hepatocyte growth factor, a c-Met receptor ligand, resulted in a significant transcriptional upregulation of COMET expression. COMET expression was additionally induced in vitro in response to hypoxic conditions; PTC cell lines grown in hypoxic conditions had a 2-fold increase in COMET expression when compared to the cell lines grown in normoxic conditions [95]. It has been reported thoroughly in the literature that c-MET receptor activation is a common mechanism exploited by cancer cells to resist treatment with vemurafenib [96]. Vemurafenib functions as an inhibitor of mutant B-Raf protein [97]. Esposito et al. thus explored a potential role for COMET expression in vemurafenib responsiveness in PTC tumor cells. It was reported that in vitro transcriptional repression of COMET in mutated BRAFV600E PTC cell line, BCPAP, had a significant increase in response to vemurafenib. Esposito et al. thus reported COMET as a potential therapeutic adjuvant for BRAF mutated PTC tumors [95].

CASC2

It is documented that LN metastasis, in general, is speculated to be associated with the dysregulated expression of lncRNA molecules [98]. The lncRNA CASC2 (cancer susceptibility candidate 2) was investigated by Zhou et al. and was found to be under expressed in PTC tissue vs. tissue from patients with nodular goiter, as well as plasma samples from the same patient cohort. This study reported that the decreased expression of CASC2 in the plasma was statistically correlated with LN metastasis in the PTC cohort. When CASC2 was overexpressed in vitro, it was found that PTC cells had decreased growth rate, as well as decreased migratory and invasive capabilities. Further evaluation confirmed that these phenotypic alterations that CASC2 has on PTC cells is through the carcinogenic modulation of E-Cadherin (decreases), as well as ZEB1 (Zinc finger E-box-binding homeobox 1) and N-Cadherin (increases), giving the cells more migratory, mesenchymal phenotypes [99].

MFSD4A-AS1

Liu et al. sought to identify specific lncRNA molecules that are associated with LN metastasis. MFSD4A-AS1 was identified from a PTC dataset cohort provided by The Cancer Genome Atlas as a significantly upregulated transcript in PTC tissue. MFSD4A-AS1 was reported to promote lymphangiogenic formation, thus enhancing the invasive propensity of PTC cells. Further exemplifying the heterogeneous outcomes of the ceRNA axis, MFSD4A-AS1 functions as a negative regulator of miR-30c-2-3p, miR-145-3p, and miR-139-5p. These miRNAs function to regulate and inhibit vascular endothelial growth factors (VEGF) A and C, thus, high expression levels enable aberrant cellular growth. Further, another study reported a significant role of VEGF-C in recurrent PTC [100]. Taken together, MFSD4A-AS1 was confirmed to serve as a putative therapeutic for anti-lymphatic metastatic PTC [98].

### 8.2. Follicular Thyroid Cancer

The molecular characterization as well as diagnostic efficiency in FTC is limited. The shared follicular cell differentiation status between FTC and PTC is still accompanied with significant variability. FTC is significantly more difficult to diagnose, due to the insufficient ability to distinguish amongst the varied follicular patterned lesions on the thyroid from true FTC tumors. Efficient diagnostic strategies for PTC are not well-suited for FTC, placing a significant need for additional diagnostic strategies [101]. The detection of biomarkers that can aid in diagnostic and prognostic strategies can potentially enable early detection and aid in differential diagnoses, significantly improving patient outcomes.

HCP5

lncRNA HCP5 (HLA complex P5) has previously served as a genetic locus marker for other thyroid diseases [102]. However, HCP5 is additionally an overexpressed transcript in FTC. Liang et al. reported a role for this molecule in the propagation and driving of proliferation and migratory propensity of FTC cells, as well as a role in angiogenesis and tumorigenesis. Further co-expression analysis via sequencing revealed a correlation between HCP5 expression and ST6GAL2 expression (alpha-2, 6-sialyltransferase 2). Liang et al. hypothesized that this expression correlation was a direct result of HCP5 upregulation of ST6GAL2, driving FTC carcinogenesis; this places a function for HCP5 as a ceRNA molecules. As of 2018, there were no annotated ceRNAs in FTC, making HCP5 the first documented [102]. HCP5 was found to act as a molecular sponge for three miRNA molecules that notably negatively regulate ST6GAL2 (miE-22-3p, miR-186-5p, miR-216a-5p). STs are notable glycotransferases that when abnormally expressed, can promote a multitude of pathologies, including cancer. These proteins can drive cell growth, chemosensitivity, and invasion in several different cancers [103,104]. To further evaluate this premise, it was also reported that expression of HCP5 and ST6GAL2 was even higher in cell lines that represent more invasive FTC when compared to cells that represent poorly invasive FTC. The role that HCP5 has on FTC carcinogenicity places this lncRNA as a potential biomarker [101].

H19

A study conducted by Dai et al. evaluated the effect of H19 downregulation observed in MI-FTC patient tissue. H19 expression was inversely correlated with tumor size and vascular invasion, as well as distant metastasis (i.e., low H19 expression, increased aggressive carcinoma phenotype). This study also reported that low H19 levels in PTC was associated with ETE and poorer disease-free outcomes; thus, further evaluation of H19 under expression in FTC was warranted, in efforts to mark this lncRNA as a potential prognostic factor of invasive FTC [33]. Stated above, distant metastasis in FTC is the most significant prognostic factor, and correlation of low H19 expression with increased distant metastasis in this study cohort is notably valuable.

Further, an additional study conducted by Xu et al. also reported low H19 expression in their FTC patient cohort. This study reported that the downregulation of H19 expression was due to its hypermethylated promoter region, and DNA methyltransferase expression was increased in FTC tissues that had low H19 expression. Xu et al. evaluated the cellular role of H19 expression in vitro and confirmed that increasing H19 expression in FTC cells ameliorated several central metastatic mechanisms, and a reduction in DNA methyltransferase expression consequently led to an increase in H19 expression. In support of this finding, increased H19 expression led to an upregulation of tumor suppressive pathways via signal decoy and ceRNA mechanisms of action, depending on its nuclear or cytoplasmic location (respectively). As a signal decoy, H19 functioned to recruit transcription factor Pax5, which lead to the activation of SOCS3 (suppressors of cytokine signaling 3) and the inhibition of the IGF1 (insulin growth factor 1) pathway. The IGF1 pathway is a characteristic tumor promoter; high IG1 levels have been positively correlated to the establishment of primary cancers, such as breast and colorectal cancer [105]. IGF2BP1 (IGF binding protein) functions as a negative regulator of IGF1, and its expression is elevated in FTC tissue. As a ceRNA molecule, H19 functions to sponge miR-29b-3p, which itself, serves as a negative regulator of IGF2BP1 mRNA. In FTC tissue, miR-29b-3p levels are elevated because of H19 under expression, and IGF2BP1 levels are consequently reduced, causing the pathological increase in IGF1 expression, driving FTC carcinogenesis [106].

GAS5

Liu et al. reported a downregulation of lncRNA GAS5 (growth arrest specific transcript 5) expression as well as an upregulation of miR-221-3p expression in FTC tissues and cells. GAS5 is described as a tumor suppressor and its loss of function strongly contributes to pathological cell growth. Experimental assessment of GAS5 and miR-221-3p mechanisms in vitro confirmed a significant role in cell cycle control and the pathological induction of cell proliferation when under and over expressed, respectively. When GAS5 expression was increased in vitro, miR-221-3p expression increased, leading to cell cycle arrest at the G0/G1 phase—significantly contributing to the immortalization of FTC cells [107].

## 9. Conclusions

The identification of differentially expressed lncRNAs provides significant opportunity for research progression in cancer [108]. Cancer cells are undoubtedly expending tremendous energy to transcribe molecular transcripts that lack functional abilities—identifying these RNAs as well as their interactive targets provides both mechanistic knowledge of carcinogenesis, as well as additional marker discovery [109]. Early diagnosis is a common need across cancer types—harnessing cancer cell growth as well as prognostic prediction significantly contributes to therapeutic options and selection [110]. Despite the vast differences in carcinogenic phenotypes, clinical presentation, and survival status, ATC, PTC, and FTC, present with relatively similar mutational burdens. This further drives the need for the identification of differentially expressed wild-type transcripts that are products of mutational burden and/or cancer cell needs. Further, ATC is therapeutic-resistant—recent studies in other cancer types have shown that ncRNA targeting in conjunction with therapy has increased responsiveness and success [111]. The application of identifying ncRNAs in cancer as carcinogenic drivers—whether through increasing metastatic phenotypes or conferring therapeutic resistance—can be used across all cancer types. Although biopsy sequencing for genomic studies has become a more common practice, this still limits the seamless identification of molecular markers. However, other cancer types have had success in identifying cancer-associated lncRNAs in the body fluids of patients—greatly expanding the feasibility of their use as biomarkers of various clinical pathological features [112].

ATC comprises 1–2% of all diagnosed TC cases, but it accounts for roughly 50% of all TC deaths, owing to its aggressive phenotype. As an exceedingly rare malignancy, the abundance of research related to improving diagnosis, prognosis, and therapeutic efficacy in other more common cancers is lacking. Due to its ability to grow significantly within a matter of weeks, there is an urgent need for better detection, early diagnosis, prevention, and improved therapeutic intervention. ATC has a 5-year OS of 0–10%, while PTC 5-year OS is 99%, and yet these two TC subsets commonly acquire the same driver mutations that propagate carcinogenesis. Better understanding of the additional factors fine-tuning genomic expression in ATC that create its hostile and lethal nature are an urgent need. Specific upregulation of lncRNAs (H19, MALAT1, HOTAIRM1, UCA1, MANCR, NEAT1) in ATC have been shown to contribute to its proliferative capacity, invasiveness, migratory propensity, and induction of EMT—all of which greatly impact the inability to control ATC in patients (Figure 1). This overexpression may also correlate with advanced TNM stage (NEAT1), LN and distant metastasis (NEAT1), and worse prognosis and OS (HOTAIRM1). Additionally, inhibition of autophagy and apoptosis mechanisms by lncRNAs (H19, MALAT1) allow for replicative immortality in these aggressive tumors. The immunosuppressive TME of ATC contributes to the inability of the immune system to fight this disease. Thus, understanding the mechanisms by which lncRNAs (UCA1) increase immune checkpoint expression and block essential cytotoxic immune cell activity can lead to new insights in the treatment of ATC through immunotherapy. Through direct interactions with known tumor suppressing miRNAs (MALAT1, HOTAIRM1), miRNA sponging (NEAT1, UCA1), and competition with miRNAs for mRNA binding (UCA1), lncRNAs drive oncogenesis in ATC via these dynamic interactions. Importantly, discovery of lncRNAs related to chemoresistance (NEAT1, H19) gives great insight into the mechanisms underlying the inability to properly treat ATC. Further progress in this area could contribute significantly to the ability to improve patient outcomes in the future. Finally, downregulation of lncRNAs (PAR5) can also lead to hyperactive signaling and uncontrolled oncogenesis via a lack of negative regulatory feedback. Further elucidation of the positive and negative control exerted by up- and downregulated lncRNAs may be an essential component in identifying biomarkers for earlier detection and predicted therapeutic responsiveness, as well as actionable targets to improve patient outcomes.

PTC accounts for roughly 80% of TC cases and represents one of the most common endocrine malignancies in young women under the age of 25. Despite the successful outcome of treatment intervention, disease recurrence remains a prominent issue and aggressive cases persist. Additionally, despite the decline in cancer incidence for many cancer types over the last 30 years, PTC incidence is still increasing. Identification of biomarkers that can aid in prognostic prediction in early cases, as well as markers that can serve as treatment targets, will contribute immensely to the prevention of disease recurrence. The molecular characterization of PTC establishment and progression is also of tremendous value to understanding other, more aggressive, TC subsets. Upregulated lncRNA transcripts that function as oncogenes (SOX2O, DANCR, lnc-OMD-1, LUCAT1, HOTTIP, H19, COMET, MFSD4A-AS1), have been identified to contribute extensively to carcinogenesis and tumorigenicity of PTC. Downregulated lncRNA transcripts (BANCR, H19, CASC2) have also been shown to promote carcinogenic behavior (Figure 1).

FTC is the second most diagnosed TC subtype, following PTC. FTC has a higher metastatic propensity when compared the PTC, which is evidenced by the fact that metastasis is present in 7–23% of cases at diagnosis. Differentiating FTC from other thyroid lesions is a large challenged faced at the time of diagnosis, therefore, genetic markers that can aid in diagnostic accuracy and efficiency is a critical unmet need. Overexpressed lncRNA HCP5, as well as underexpressed H19 and GAS5 serve thus far as potential biomarkers for FTC diagnosis (Figure 1). Extensive evaluation of additional lncRNA molecules in FTC is warranted.

The majority of annotated lncRNAs in TC focus on their roles as molecular sponges—impacting transcriptomic and genomic expression levels and cellular behavior. Additional evaluation of other functional lncRNA roles, such as protein scaffolds and signal decoys, is extensively warranted. Use of in silico data is indispensable when identifying transcripts that are differentially expressed and is an invaluable tool for cancer research and elucidation of cancer cell molecular phenotypes.

Cancer arises from a series of genetic alterations that drive carcinogenesis. Thus, TC represents a diverse population of genetic markers, mutations, phenotypes, establishment, and progression. Identification of markers specific to well-differentiated and undifferentiated TC, as well as identification of overlapping markers that function in similar or contradicting ways in these TC subsets, will contribute significantly to understanding the vast amount of unknown molecular function of carcinomas arising in the thyroid. lncRNA molecules lie at the interface of cellular function, giving tremendous insight into how cells re-program into transformed phenotypes. The versatility and tissue-specificity of lncRNA expression makes them excellent candidates for genomic and transcriptomic evaluation. RNAi-based therapeutics has thus far shown excellent potential in the targeted delivery of gene silencing machinery into tumors [113]. lncRNA study in TC has made excellent progress thus far, and this review highlights some of the recent advances and successes in RNA biology. Further functional evaluation of differentially expressed lncRNAs will enable a better understanding of thyroid carcinoma complexity.

## Figures and Tables

**Figure 1 biology-13-00304-f001:**
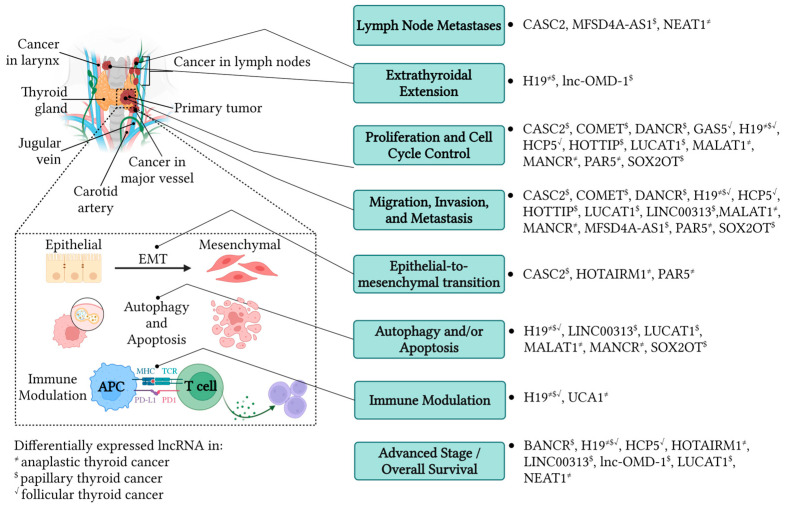
Summary of functional roles of differentially expressed long non-coding RNAs in ATC, PTC, and FTC discussed in this review [Figure created on BioRender.com].

**Table 1 biology-13-00304-t001:** Summary of differentially expressed long non-coding RNAs in ATC, PTC, and FTC discussed in this review. Grouped by cancer subtype; table shows cancer subtype, lncRNA name, up- or downregulation status in ATC or PTC, selected roles in ATC or PTC, from left to right.

Thyroid Cancer Subtype	lncRNA Name	Expression Pattern	Role in Thyroid Cancer
Anaplastic Thyroid Cancer	MALAT1	↑	miR-200p-3a repression and FOXA1 activation impacting proliferation, migration/invasion, autophagy, and apoptosis
	HOTAIRM1	↑	Correlated with decreased patient survival; prevents functional miR-144 and induces EMT
	UCA1	↑	Drives aberrant c-myc expression and oncogenesis [49]; sponges miR-148a, increases PD-L1 expression, and CTL activity
	MANCR	↑	Drives apoptotic inhibition and proliferative and invasive induction
	NEAT1	↑	Associated with advanced TNM stage, LN metastasis, and chemoresistance [54]; inhibits miR-9-5p; drives hypoxia-induced carcinogenic behavior
	PAR5	↓	Unable to inhibit EZH2, leading to its overexpression and increased oncogenic activity
	H19	↑ ^≠^	Influences proliferation, colony-forming capacity, invasive potential, migratory propensity, and apoptotic induction
Papillary Thyroid Cancer	H19	↑↓ ^≠^	Conflicting roles: low levels may promote carcinogenesis and modulate TME activity; high levels may serve as an oncogene
	SOX2OT	↑	Biomarker in PTC; correlated with cancer onset and progression
	DANCR	↑	Biomarker in PTC; correlated with cancer onset
	TINCR	=	Reported biomarker in PTC; associated with various clinicopathological features of PTC
	lnc-OMD-1	↑ *	Significantly correlated with ETE and more aggressive PTC
	LUCAT1	↑	Correlated with advanced TNM stage; impacts G1 cell cycle control (p21/p53 axis), proliferation, invasion, and apoptotic prevention
	HOTTIP	↑	Promotes cellular proliferation and metastasis via miR-637 repression and Akt1 activation; correlates with tumor volume
	LINC00313	↑	Correlated with worse prognosis; promotes migration and clonogenicity; inhibits apoptosis; downregulates miR-4429
	BANCR	↓ *	Low levels correlated with advanced TNM staging and poorly differentiated PTC
	COMET	↑	Oncogenic driver in PTC (via MET/MAPK activation) leading to cellular proliferation, survival, and metastasis
	CASC2	↓	Correlated with LN metastasis; associated with growth rate, migratory and invasive capabilities, and EMT
	MFSD4A-AS1	↑	Correlated with LN metastasis; promotes lymphangiogenic formation, and enhances PTC cell invasiveness
Follicular Thyroid Cancer	HCP5	↑	Drives proliferation, migration, angiogenesis, and tumorigenesis in FTC; correlated with more invasive FTC
	H19	↓ ^≠^	Inversely correlated with tumor size, vascular invasion, and distant metastases; associated with ETE and likelihood of recurrence
	GAS5	↓	Regulates G0/G1 cell cycle control and cellular proliferation

^≠^ Known to be differentially expressed in ATC, PTC, and FTC; * Low expression level found specifically in higher-stage PTC.

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
