# Peer review of "Long Non-Coding RNAs as Determinants of Thyroid Cancer Phenotypes: Investigating Differential Gene Expression Patterns and Novel Biomarker Discovery"

_biology, 2024, doi:10.3390/biology13050304_

Round 1

Reviewer 1 Report (New Reviewer)

Comments and Suggestions for Authors

The manuscript titled "Long Non-Coding RNAs as Determinants of Thyroid Cancer Phenotypes: Investigating Differential Gene Expression Patterns and Novel Biomarker Discovery" provides an extensive review of the role of long non-coding RNAs (lncRNAs) in thyroid cancer. The manuscript addresses a crucial and timely topic within cancer research, focusing on thyroid cancer. It highlights the significance of lncRNAs in determining cancer phenotypes, which is a relatively underexplored area with potential implications for diagnosis, prognosis, and therapy.

While the manuscript thoroughly covers lncRNAs' roles in thyroid cancer, it could further benefit from a discussion on the challenges of translating this knowledge into clinical practice. Addressing the gaps between research findings and their application in diagnostics or therapy could provide a more rounded perspective. Moreover, given the rapid evolution of the field, it would be beneficial to include a section on emerging technologies and methodologies in studying lncRNAs, which could open new research avenues or improve current understanding.

Author Response

Reviewer #1:

Comment: While the manuscript thoroughly covers lncRNAs' roles in thyroid cancer, it could further benefit from a discussion on the challenges of translating this knowledge into clinical practice. Addressing the gaps between research findings and their application in diagnostics or therapy could provide a more rounded perspective. Moreover, given the rapid evolution of the field, it would be beneficial to include a section on emerging technologies and methodologies in studying lncRNAs, which could open new research avenues or improve current understanding

Response: We agree that these topics are immensely important and would add significant value to the paper. In our revision, we addressed the translation into clinical practice, application in diagnostics/ therapy, emerging technologies, and methodologies for evaluating lncRNAs. Thank you for these very helpful suggestions.

Reviewer 2 Report (New Reviewer)

Comments and Suggestions for Authors

Long non coding RNAs being a novel category of biomarker and thyroid cancer the most common endocrine cancer, the topic of this review is quite significant. The article is written well with a description of each thyroid cancer type and the lncRNAs identified. A few comments to be considered are given below.

1.    Some of the articles on the long non-coding RNAs in thyroid cancer were not cited. Please include or explain why these articles were not cited. Please mention the lncRNAs such as SOX2OT, DANCR and FAL1 (other than the ones explained in detail) so far reported at the end of each type of thyroid cancer.

(a)    Wang H. LINC00092 Enhances LPP Expression to Repress Thyroid Cancer Development via Sponging miR-542-3p. Horm Metab Res. 2024 Feb;56(2):150-158.

(b)   Icduygu FM et.al.,Expression of SOX2OTDANCR and TINCR long non‑coding RNAs in papillary thyroid cancer and its effects on clinicopathological features. Mol Med Rep. 2022 Apr;25(4):120. doi: 10.3892/mmr.2022.12636.

(c)    Tang J, et.al., LncRNA DANCR upregulates PI3K/AKT signaling through activating serine phosphorylation of RXRA. Cell Death Dis. 2018 Dec 5;9(12):1167. doi: 10.1038/s41419-018-1220-7.

(d)   Jeong S, et.al.,. Simultaneous Expression of Long Non-Coding RNA FAL1 and Extracellular Matrix Protein 1 Defines Tumour Behaviour in Young Patients with Papillary Thyroid Cancer. Cancers (Basel). 2021 Jun 28;13(13):3223. doi: 10.3390/cancers13133223.

2.    Please include the lncRNA studies on other thyroid cancer types such as Hurthle cell cancer and Medullary thyroid cancer (Adam et.al., Endocr Pathol, 2017 Sep;28(3):207-212, Luzon et.al., Orphanet J Rare Dis. 2021 Jan 6;16(1):4, Chu et.al., Exp Mol Pathol. 2017 Oct;103(2):229-236).

3.    Line 76-79, “Most TC cases present, genetically, with driver mutations in the MAPK (mitogen activated protein kinase) pathway, which both propagates and progresses uncontrolled cell proliferation. Mutations in this pathway include primarily mutated BRAF and RAS, significantly contributing to the constitutive activation of this highly proliferative signal transduction cascade”. Does ‘both’ indicates mutations in BRAF and RAS in line 76? Please clarify.

4.    Line, 127, Please include a short introduction paragraph on types of Thyroid cancer- undifferentiated and differentiated, incidence in males/females and the available statistics before the Section 2.

5.    Line, 141, Section 3, please include a short introduction on differentiated thyroid cancer and types briefly.

6.    Line 226, please include an introduction to lncRNAs in different cancers. Li et.al, Cancer Res. 2015 Sep 15; 75(18): 3728–3737 has described about an open platform to explore the function lncRNAs in different cancers. Please include this article in the review, as it is reported that 1900 lncRNAs in thyroid cancer were identified. Using recent large-scale RNA-seq datasets such as The Cancer Genome Atlas (TCGA), “The Atlas of Noncoding RNAs in Cancer” has been developed (TANRIC, 

http://bioinformatics.mdanderson.org/main/TANRIC:Overview). It would be worthwhile to mention this database and related recent articles in the introduction.

7.    In Table 1, it would be clearer and easier for the readers, if the titles of each column are mentioned on the top rather than mentioning them in the table legend. Please include the statistical data available in each report.

8.    In Figure 1, please denote lncRNA identified in each type of thyroid cancer (by superscript numbers or symbols and footnote). This would help the readers to understand the lncRNAs identified in each thyroid cancer type, from the figure itself.

9.    In conclusion, the authors have summarized the previous sections. Please elaborate the gaps in the studies reported so far especially in the differential gene expression analysis and also the future directions and approach to be taken to identify novel lncRNAs as diagnostic and therapeutic targets.

Author Response

Reviewer #2:

  1. Some of the articles on the long non-coding RNAs in thyroid cancer were not cited. Please include or explain why these articles were not cited. Please mention the lncRNAs such as SOX2OT, DANCR and FAL1 (other than the ones explained in detail) so far reported at the end of each type of thyroid cancer.

(a)    Wang H. LINC00092 Enhances LPP Expression to Repress Thyroid Cancer Development via Sponging miR-542-3p. Horm Metab Res. 2024 Feb;56(2):150-158.

(b)   Icduygu FM et.al.,Expression of SOX2OTDANCR and TINCR long non‑coding RNAs in papillary thyroid cancer and its effects on clinicopathological features. Mol Med Rep. 2022 Apr;25(4):120. doi: 10.3892/mmr.2022.12636.

(c)    Tang J, et.al., LncRNA DANCR upregulates PI3K/AKT signaling through activating serine phosphorylation of RXRA. Cell Death Dis. 2018 Dec 5;9(12):1167. doi: 10.1038/s41419-018-1220-7.

(d)   Jeong S, et.al.,. Simultaneous Expression of Long Non-Coding RNA FAL1 and Extracellular Matrix Protein 1 Defines Tumour Behaviour in Young Patients with Papillary Thyroid Cancer. Cancers (Basel). 2021 Jun 28;13(13):3223. doi: 10.3390/cancers13133223.

Comment: Thank you for your suggestion, we included information regarding DANCR, FAL1, and SOX20T in our revision—Wang et al., Icduygu et al., and Tang et al. – thank you for these valuable sources and papers

  1. Please include the lncRNA studies on other thyroid cancer types such as Hurthle cell cancer and Medullary thyroid cancer (Adam et.al., Endocr Pathol, 2017 Sep;28(3):207-212, Luzon et.al., Orphanet J Rare Dis. 2021 Jan 6;16(1):4, Chu et.al.,Exp Mol Pathol. 2017 Oct;103(2):229-236).
    1. Comment: When this paper was initially submitted, it was agreed to discuss ATC, PTC, and then we revised and added FTC. We believe medullary thyroid cancer is an extremely important subtype, and as such, would likely deserve a separate paper in its entirety, and was omitted in order to keep the consistency in our paper.

  1. Line 76-79, “Most TC cases present, genetically, with driver mutations in the MAPK (mitogen activated protein kinase) pathway, which bothpropagates and progresses uncontrolled cell proliferation. Mutations in this pathway include primarily mutated BRAF and RAS, significantly contributing to the constitutive activation of this highly proliferative signal transduction cascade”. Does ‘both’ indicates mutations in BRAF and RAS in line 76? Please clarify
    1. Comment: Sentence was rectified for clarity.

  1. Line, 127, Please include a short introduction paragraph on types of Thyroid cancer- undifferentiated and differentiated, incidence in males/females and the available statistics before the Section 2.
    1. Comment: This information was added to our revision. Thank you.

  1. Line, 141, Section 3, please include a short introduction on differentiated thyroid cancer and types briefly.
    1. Comment: This information was added to our revision. Thank you.

  1. Line 226, please include an introduction to lncRNAs in different cancers. Li et.al, Cancer Res. 2015 Sep 15; 75(18): 3728–3737 has described about an open platform to explore the function lncRNAs in different cancers. Please include this article in the review, as it is reported that 1900 lncRNAs in thyroid cancer were identified. Using recent large-scale RNA-seq datasets such as The Cancer Genome Atlas (TCGA), “The Atlas of Noncoding RNAs in Cancer” has been developed (TANRIC, 

http://bioinformatics.mdanderson.org/main/TANRIC:Overview). It would be worthwhile to mention this database and related recent articles in the introduction.

  1. Comment: This information was added to our revision. Thank you. We, ourselves use this resource, and appreciate the reminder for its addition to the paper.

  1. In Table 1, it would be clearer and easier for the readers, if the titles of each column are mentioned on the top rather than mentioning them in the table legend. Please include the statistical data available in each report.
    1. Comment: rectified for clarity
  2. In Figure 1, please denote lncRNA identified in each type of thyroid cancer (by superscript numbers or symbols and footnote). This would help the readers to understand the lncRNAs identified in each thyroid cancer type, from the figure itself.
    1. Comment: rectified for clarity
  3. In conclusion, the authors have summarized the previous sections. Please elaborate the gaps in the studies reported so far especially in the differential gene expression analysis and also the future directions and approach to be taken to identify novel lncRNAs as diagnostic and therapeutic targets.
    1. Comment: Thank you for this suggestion. We added this to our revision and agree it added tremendous value to the conclusion.

Reviewer 3 Report (New Reviewer)

Comments and Suggestions for Authors

            COMMENTS  

The manuscript titled “Long Non-Coding RNAs as Determinants of Thyroid Cancer Phenotypes: Investigating Differential Gene Expression Patterns and Novel Biomarker Discovery” of Nicole R. DeSouza et al., reports a literature review of data concerning the expressions of non-coding RNAs in thyroid cancers.

Summary:

Lines 11-13: “depending on how similar the cancer cell resembles its original thyroid cell form”. This is a misguided sentence because of morphological features of epithelial follicular cells of papillary thyroid carcinoma. In fact, these cells show specific features such as nuclear grooves, nuclear enlargement by elongation and overlapping, chromatin clearing, margination and glassy nuclei. However, these morphological alterations do not affect the prognosis because the most of PTC have a really good prognosis.

Abstract:

Abstract section is adequately describing this study.

Introduction:     

This section is adequately describing the aims of study.

Anaplastic Thyroid Cancer, Differentiated Thyroid Cancer, Anaplastic Thyroid Cancer vs. Well-Differentiated Thyroid Cancers (Papillary Thyroid Cancer and Follicular Thyroid Cancer), Current Thyroid Cancer Treatment Modalities, lncRNAs in Anaplastic Thyroid Cancer, lncRNAs in Well-Differentiated Thyroid Cancer   : 

These sections provide sufficient information.

Conclusions:      

The conclusions are relevant.

Table and Figures: give a helpful visual representation of study.

References:

References are adequate.

Decision:

This paper is well-written and may be accepted for publication after minor revision.

Author Response

Reviewer #3:

Lines 11-13: “depending on how similar the cancer cell resembles its original thyroid cell form”. This is a misguided sentence because of morphological features of epithelial follicular cells of papillary thyroid carcinoma. In fact, these cells show specific features such as nuclear grooves, nuclear enlargement by elongation and overlapping, chromatin clearing, margination and glassy nuclei. However, these morphological alterations do not affect the prognosis because the most of PTC have a really good prognosis.

Comment: Thank you for your revision comments and feedback! This sentence was removed for lack of information clarity.

This manuscript is a resubmission of an earlier submission. The following is a list of the peer review reports and author responses from that submission.

Round 1

Reviewer 1 Report

Comments and Suggestions for Authors

In this paper, DeSouza et al. tried to write a review about lncRNAs in Thyroid cancer. There are many reviews on the matter and this one does not contribute in any way to the field.

1. The lack of references is flagrant. No references but one on the thyroid cancer introduction. The table 1 should have references too. And, checking older reviews about lncRNAs, it could be easily notice that a lot of important articles on the matter are lacking. Check as example: Long noncoding RNAs: emerging players in thyroid cancer pathogenesis (2018) or a most recent one The Role of Long Non-Coding RNAs in Thyroid Cancer (2020). It is not enough a review only with 34 references!!

2. The classification of TC is not explained, the relationship of the different subtypes and driver mutations is also not explained. The authors focus only in PTC and ATC, without noticing that there are other important TC subtypes like FTC. Explain why do you focus only in those. No references about mutational burden, clinical stages, treatment, etc. on ATC, PTC.

3. Figure 1 has incomplete sentences, poor quality.

4. The title says therapeutic target, but nothing about therapy is discussed on the paper.

5. Line 15, PTC arises in women younger than 25 years old!!! That’s not true, at all.

Comments on the Quality of English Language

6. Line 96-98, really convoluted sentence.

7. c-myc or c-Myc??

8. Line 124, disproportionately?? Please be more scientific.

9. Line 138-139, it implies that there are more lncRNA than H19 that have different behavior between ATC and PTC, you are generalizing from one example. 

Author Response

Please see attached: Response to reviewer #1

Reviewer 2 Report

Comments and Suggestions for Authors

I have thoroughly reviewed the article and find it to be a valuable contribution to the study of long non-coding RNAs (lncRNAs) in the context of thyroid cancer phenotypes and providing suggested comments to enhance the quality of the paper:

1. The "Simple Summary" section should be concise and focused on highlighting the key points of the article. Consider revising this section to provide a brief and easily understandable overview of the study's main findings and contributions. Avoid unnecessary details and prioritize clarity and accessibility to engage a broader audience. By streamlining this section, the article will effectively convey its significance to readers, increasing its overall impact and accessibility.

2. Strengthen the Introduction: The introduction lacks a coherent and logical progression of ideas. There is a need to establish clear relationships between different paragraphs to help guide the reader's understanding of the context and significance of your research. Try to construct a narrative that starts with a broad context, slowly narrows down to your specific research question. Ensure that your thesis statement is clear and positioned at the end of the introduction to provide a roadmap for the rest of the paper. It is crucial that this section hooks the reader's interest and clearly states the purpose of your study. Consider a thorough revision of this section for clarity and flow.

3. In lines 58-60, there is no compelling reason to exclude the investigation of follicular thyroid cancer, and it is necessary to add a subtitle to this section in the article.

4. In lines 65-70: Consider removing lines 65-70 to maintain the clarity and coherence of the article, ensuring the reader stays on track with the main points.

5.Figure 1 needs improvement as it lacks clarity and has been sourced from an external website, which not be directly relevant to the main focus of the article on lncRNA and thyroid cancer. Consider replacing Figure 1 with a more relevant and focused figure that illustrates the specific effects of lncRNA on the thyroid. This will enhance the understanding of the readers regarding the main subject matter and strengthen the connection between the content and your research question.

6.Figure 2 seems to restate fundamental concepts without providing any meaningful insight into the understanding of the article. It would be beneficial to replace Figure 2 with a more informative and relevant illustration that directly aids in comprehending the content. This will enhance the overall quality of the paper and reinforce the presentation of your research findings. When including figures, ensure that the emphasis is on lncRNA and its direct relevance to thyroid cancer. The figures should visually represent the relationship between lncRNA and the mechanisms involved in thyroid cancer development or progression. By concentrating on this aspect, the figures will effectively complement and reinforce the key points of your research, providing a more comprehensive understanding of the study's primary subject matter.

7.In line 135-169: The content from lines 135 to 169 appears to be distracting and disrupts the coherence of the article. It confuses the reader and deviates from the main objective of the study. To maintain a clear and focused narrative, it is recommended to remove this section entirely. By doing so, the article's message will be more concise and effectively convey the research findings and conclusions without unnecessary distractions.

8.Sections 6 and 7 require significant improvement as they are overly tedious and repetitive, solely relying on citing other researchers' work without providing any novel insights or data. Consider reorganizing these sections to highlight the key findings from other studies concisely, focusing on the relevance to your research. Instead of long paragraphs, use clear and concise sentences to present the information effectively. Additionally, ensure that these sections complement and contribute to the overall narrative of the article, providing readers with valuable context and knowledge without overwhelming them with excessive citations. By revising these sections, the article will become more engaging and impactful, making it a more valuable contribution to the field.

Comments on the Quality of English Language

From a grammatical standpoint, the quality of the article's writing appears to be appropriate. there are no significant grammatical errors that hinder comprehension. However, to enhance the overall clarity and coherence of the paper, you may want to focus on the organization and flow of the content, as mentioned in the previous comments. Ensuring that the introduction, methods, results, and conclusion are logically connected and smoothly transitioned will further improve the overall readability and impact of the article.

Author Response

Please see attached: Response to reviewer #2

Round 2

Reviewer 1 Report

Comments and Suggestions for Authors

The paper have improve profusely, undoubtedly. However, there are issue that have been brought up but both reviewers and have not been adressed.

1. As reviewer 2 said: there is no compelling reason to exclude follicular thyroid cancer. Or at least, restructure the tittle and introduction as a comparison of these two types, that are the contrasting points of the spectrum of thyroid cancer.

2. The figure 1 have not improve, nor by my comments or the other reviewer's suggestions. As an example. Figure 1 says: Median overall survival of 99% for regional disease.

Author Response

We kindly thank you for your updated revisions-

  1. We added a section on FTC
  2. Sentences were corrected on Figure 1 

All the best, 

Jan Geliebter

Nicole DeSouza

Round 3

Reviewer 1 Report

Comments and Suggestions for Authors

The sentences that should be corrected on Figure 1 are still on the figure, not on the foot note. SInce you added FTC, please, erase the polemic Figure 1, since is not really necessary.

Author Response

The sentences that should be corrected on Figure 1 are still on the figure, not on the foot note. Since you added FTC, please, erase the polemic Figure 1, since is not really necessary.

We removed figure 1 and upon review on our end, believe it does not contribute to this review paper.